# CNN-Based Target Recognition and Identification for Infrared Imaging in Defense Systems

**DOI:** 10.3390/s19092040

**Published:** 2019-04-30

**Authors:** Antoine d’Acremont, Ronan Fablet, Alexandre Baussard, Guillaume Quin

**Affiliations:** 1ENSTA-Bretagne, UMR 6285 labSTICC, 29806 Brest, France; alexandre.baussard@utt.fr; 2MBDA France, 92350 Le Plessis-Robinson, France; guillaume.quin@mbda-systems.com; 3Institut Mines-Télécom, UMR 6285 labSTICC, 29238 Brest, France; ronan.fablet@imt-atlantique.fr

**Keywords:** deep learning, CNN, target identification and recognition, infrared imaging

## Abstract

Convolutional neural networks (CNNs) have rapidly become the state-of-the-art models for image classification applications. They usually require large groundtruthed datasets for training. Here, we address object identification and recognition in the wild for infrared (IR) imaging in defense applications, where no such large-scale dataset is available. With a focus on robustness issues, especially viewpoint invariance, we introduce a compact and fully convolutional CNN architecture with global average pooling. We show that this model trained from realistic simulation datasets reaches a state-of-the-art performance compared with other CNNs with no data augmentation and fine-tuning steps. We also demonstrate a significant improvement in the robustness to viewpoint changes with respect to an operational support vector machine (SVM)-based scheme.

## 1. Introduction

As shown on reference datasets such as ImageNet [1], convolutional neural networks (CNNs) have become the state-of-the-art approaches for object classification in images. With the introduction of networks capable of dealing with both detection and classification [2,3,4,5], neural networks can provide an all-in-one solution for target detection and classification. However, there are still issues remaining such as the need for CNNs of very large groundtruthed datasets or classification failures that a human eye would not make [6,7]. In this study, we address target identification and recognition in defense application contexts, as encountered in synthetic aperture radar imaging, sonar imaging, or infrared imaging. In a defense context, no large-scale real datasets can generally be collected. To overcome this problem, simulation techniques have been developed to provide realistic images. They can model viewpoint changes, object variabilities, scene clutter, etc, which are known to be necessary during training. Thus, an alternative strategy is to train the architecture using these simulated data, if realistic enough, and then test it on real data, as illustrated in Figure 1. Since a high robustness is of key importance in defense applications, modular solutions that would be easier to understand and evaluate may be preferred, for instance, with distinct modules for detection and classification. Based on this observation, we focus on modular solutions and assume that we are provided with a target detection algorithm, which extracts image patches for a recognition and identification stage.

In this paper, to address target identification and recognition in infrared images, we propose a compact and fully convolutional neural network with global average pooling (GAP), referred to as cfCNN. The proposed architecture is tested against three separate datasets with increasing quality. Benchmarking experiments on real IR images patches demonstrate the relevance of the cfCNN with respect to state-of-the-art CNN architectures and an approach based on support vector machines (SVM). They stress the importance of realistic synthetic dataset. Given the targeted operational contexts, we study the robustness of our cfCNN against possible perturbations introduced by the detection stage. To simulate such behaviour, we train our network on images with centred targets and test it on translated or scaled inputs. We compare the cfCNN performances against state-of-the-art CNN architecture and a SVM based approach. The results also highlight that Global Average Pooling improves robustness against disrupted inputs.

The rest of the paper is organized as follows. Section 2 briefly reviews related works and the description of the problem. In Section 3, we describe the datasets of simulated and real data considered in this study. The proposed CNN architecture is presented in Section 4. Section 5 presents numerical experiments, and we further discuss our key contribution in Section 6.

## 2. Related Work and Problem Statement

### 2.1. Infrared Imaging for Object Detection, Recognition, and Tracking

Infrared (IR) imagery is a conventional imaging modality for defense applications. In this domain, one can be interested, for example, in the detection of large, small, or point targets [9,10,11,12]; automatic target tracking and recognition [13,14]; or target classification and recognition. In this last field, several methods have been proposed to extract features and then to classify objects [15,16,17,18]. Specific classifiers such as nearest neighbor classifier, Bayesian classifier, or SVMs can be used [16,17]. When video sequences are available, other methods can be proposed. For example, in Reference [19], it is shown that background suppression can improve the detection stage. Indeed, successive frames can increase the robustness to background variations. However, such techniques may introduce masking on cold or stationary targets that blend in the environment.

In this paper, we propose to use a deep-learning approach for target recognition and identification. In the next part, we propose a brief overview of deep learning approaches focused on defense applications and more specifically to IR images.

### 2.2. Deep Learning for Recognition and Identification in Infrared Images

A strong interest in deep learning has emerged in recent years, especially Convolutional neural network (CNN). This class of artificial neural networks provides excellent results in various domains [20]. They are able to automatically and adaptively learn spatial hierarchies of features through a backpropagation by using multiple building blocks (convolution layers, pooling layers, fully connected layers, etc.). They outperformed previous shallow machine learning techniques in image classification tasks and could also be adapted to work on other computer vision problems such as pose estimation [21], super-resolution [22], or image segmentation [23].

Most CNN applications for object detect and recognition have been developed for optical images. Using other imaging sensors (radar, sonar, and infrared) may lead to more challenging problems due to the intrinsic image characteristics. CNNs have been successfully used for the classification of ground targets in SAR imagery [24] or in synthetic aperture sonar (SAS) imagery [25]. In this paper, we focus on mid-wavelength infrared (MWIR) imagery. Some CNNs have already been proposed for this kind of images for target classification [26,27] or for image enhancement [28,29], for example.

A recent study [30] combines a CNN for object classification with an automatic target detection (ATD) algorithm to perform Automatic Target Recognition (ATR). This approach was applied to long-wavelength infrared imagery and evaluated for different use-cases. This work highlights the relevance of a multistage ATR algorithm. It makes feasible the separate training and validation of each stage of the ATR chain. This is regarded as a key feature when dealing with real-world applications.

ATR and ATD in IR are highly dependent on the quality of the input. The infrared radiation received by the camera sensor can vary a lot depending on the meteorological conditions and sensor calibration. Among the perturbations that could affect the performance of an ATR algorithm in IR, the state of targets can also affect the performance of ATR and ATD systems. For example, cold and stationary vehicles blend with the background or plumes of engine smoke partially mask the target. In a realistic defence context, we may also expect noncooperative behaviour, challenging weather conditions, and the use of camouflage or countermeasures.

### 2.3. Learning Strategies with Simulated Datasets

In several applications and, more specifically, in a military context, only a limited amount of data is available, within which only a fraction might be fully or partially groundtruthed. For example, the target may not be identified but only labeled according to generic categories such as car, truck, etc. Advances in image simulation provide an alternative to the collection of representative image datasets as illustrated by Reference [31] for SAR imaging or the use of unsupervised learning strategies [32], transfer learning [33], and improved convolution layers [34]. Such simulation-based strategies have been investigated for applications in optical imaging. For instance, in Reference [35], a robotic arm was trained in a simulated environment involving varied lighting and colour conditions before its deployment in real situations. This process was also studied in Reference [36] to improve autonomous driving, where the system could learn driving strategies on video games before their application to real-world scenarios.

For MWIR imaging, realistic synthetic images may be simulated in various environments knowing viewpoint-related MWIR signatures of targets of interest [29]. The calibration of these MWIR signatures is expected to play a key role for the transfer of models trained on simulated data to real MWIR images.

### 2.4. Viewpoint Invariance

Object recognition has long addressed viewpoint invariance issues. Regarding CNNs, Reference [37] provides interesting insights. It supports that viewpoint invariance is mostly enforced by the last fully connected layer of a network. In addition, when using fine-tuning to improve the viewpoint invariance, only the higher layers of a network are affected. Studies on object localization using deep learning further stress the ability of CNNs to extract viewpoint-invariant image representations [38,39]. Two other features were shown to impact viewpoint-invariance properties in CNNs. Reference [40] pointed out that deeper networks may be more sensitive to object translation. Global Average Pooling (GAP) introduced by Lin et al. in Reference [41] also led to a better object recognition performance when dealing with translated inputs. Reference [42] further leveraged performance gain when using GAP for object localization.

## 3. Considered Datasets

### 3.1. Simulated and Real Infrared Data

As stated in the introduction, we address target recognition and identification on real MWIR images using CNN models trained with synthetic datasets. With the goal of evaluating how the quality of the simulation data affects the recognition and identification performances, we consider 3 different datasets, referred to as DS1, DS2, and DS3. All evaluations are carried out on a real dataset, referred to as DR.

The simulation of an IR scene requires specific tools. The databases were created using OKTAL-SE. This software includes several modules allowing for the creation of multiple signatures for vehicles and the modification of atmospheric and shooting conditions. By combining the outputs of those modules, we can create a wide variety of realistic IR scenes.

As described in Figure 2, the first step usually involves the creation of the vehicle signature using real video sequences or measurements. We then use a secondary toolset to update the signature based on the meteorological conditions and finally combine the results in a scene editor. During this final step, we can generate one or multiple databases entry using snapshots from a simulated IR sensor.

The contents of each dataset are briefly detailed below:**DS1** is composed of 17,280 images, each one containing a target. Six targets are available: two battle tanks (referenced as m1 and m2), three armoured cars (referenced as c1, c2, and c3), and one truck (t1). Thus, we have 2880 images for each target. The simulated data in DS1 were obtained using basic 3-D models of the targets and a basic model for generating the infrared signatures in the images. The resulting MIWR signatures may not be very accurate: heat transfer and diffusion are not taken into account, resulting in flat and uniform hot spots on the vehicles.**DS2** is composed of 23,040 images, each one comprising a single target. In addition to the six classes represented in dataset DS1, two other truck classes have been appended (referenced as t2 and t3). Simulated target MIWR signatures and 3-D models of the targets have been refined to provide a more realistic simulation dataset. More clearly, we took into account 2-dimensional heat transfers for each surface based on its composition. This results in smoother temperature gradients.**DS3** is composed of the same simulated scenes and targets as in dataset DS2, using target MWIR signatures recalibrated using real MIWR images. Temperature measurements on real vehicles were also used to validate or rectify physical models used in dataset DS2. As such, this dataset is the most realistic one among the three simulation datasets.**DR** contains 40,000 images extracted from infrared videos sequences. Each images has a size of 128×128 pixels. The targets available in DR are m1, m2, c1, c3, t1, and t3, meaning that no test examples will be available for classes c2 and t2.

We may point out that the simulated datasets include, for each target, images simulated for varied camera viewpoints, target positions and orientations, target state (i.e., vehicle’s engine off, vehicle’s engine on, several temperature levels of different parts of the targets like the engine or the exhaust pipe, and meteorological conditions. As such, this simulation procedure embeds data augmentation based on MIWR imaging characteristics; no additional data augmentation procedure is considered during the training.

### 3.2. Training Datasets

From the simulation datasets described in the previous section, we build training datasets for target recognition and identification as follows. Each sample in datasets DS1, DS2, and DS3 is a 640 × 512 image along with the associated target type, position, and bounding box. This bounding box is generated during the simulation process. It takes into account the position of the target and the viewpoint. Consequently, it delimits perfectly the target within the image. As illustrated in Figure 3, inputs for CNNs presented in the next Section are 128 × 128 image patches. For a given bounding box $d_{x}\times d_{y}$, we consider a squared patch of size $1.1\times \max(d_{x},d_{y})$ centered in the center of bounding box. Using a bilinear interpolation, we rescale this patch to a 128 × 128 patch. This process does not affect the shape and aspect ratio of the target within the image.

Using this procedure, we create three training target datasets referred as TRS1, TRS2, and TRS3. We randomly remove 4096 exemplars in each dataset that will be used for validation purposes. Overall, training dataset TRS1 comprises 13,184 patch images, and datasets TRS2 and TRS3 comprise 18,944 patch images.

### 3.3. Test Datasets

Three datasets are used during the experiments. The first one corresponds to the real images in dataset DR with the original bounding boxes; thus, we keep the same name. The bounding boxes are manually drawn around the target. We use this dataset to benchmark CNN architectures when trained from datasets TRS1, TRS2, or TRS3.

To evaluate the robustness of the proposed framework to target localization errors in the detection stage (i.e., when the target is not centered in the patch image), we need all target information, especially the center of the target, to be precisely known which is not the case for real data. Therefore, we build other test datasets from simulation data. We consider TRS3 dataset, which is the most realistic simulation dataset and leads to the best reccognition performance when used as training dataset. as shown in Section 5.1. We proceed as follows to generate our test dataset, referred to as TTS. We build this dataset from the 4096 images randomly removed (see Section 3.2) from dataset DS3. For a given translation direction, namely horizontal or vertical, we apply a translation of the bounding box, of which the magnitude is a predefined fraction of the dimension of the bounding box along the considered translation direction. We then proceed as described previously to extract 128 × 128 exemplars based on the initial bounding box parameters (i.e., prior to translation). We apply this procedure both for vertical and horizontal directions. For the vertical axis, the direction UP refers to a translation of the bounding box towards the top of the image and DOWN refers to that towards the bottom of the image. For the horizontal axis direction, LEFT refers to a translation of the bounding box to the left of the image and conversely for direction RIGHT. We report an example in Figure 4 for an upward translation with a factor of 0.4. We create a dataset with exemplars for translation factors from 0.1 to 0.5 for all four directions.

We create a dataset referred to as TSS to assess the robustness to scaling effects, typically related to changes in the shooting distance. Here, we rescale the bounding box with a factor ranging from 0.5 to 1.5, keeping the center of the bounding box unchanged, as shown in Figure 5. We may remind that the final patch datasets rescale the patches associated with the considered bounding boxes to 128 × 128 patches.

## 4. Proposed cfCNN

This section presents the considered CNN architecture (Section 4.1) and details training issues (Section 4.2).

### 4.1. Description of the cfCNN

Using as basis the network introduced in Reference [30], we further rely on the recent findings reported in Reference [43] to design a fully convolutional network. This allows us to reduce the number of parameters from 310 thousand in Reference [30] to approximately 130 thousand. The number, the size of the filters, and the type of operations included in the network are detailed below.

Figure 6 and Table 1 describe the overall structure of the network, with 7 convolution layers, 3 max pooling operations, and a global average pooling operation preceding the Softmax layer. After each max pooling layer, the number of filters is doubled. All units in these layers uses the Leaky ReLU [44] nonlinearity instead of the standard ReLU. By adding a small slope for negative inputs, Leaky ReLU is one possible solution to prevent vanishing gradient issues during back propagation. This is known to help against dying units during training. This choice differs from the architectures presented in Reference [24,26,30]. It was shown to be particularly relevant for the transfer of the trained network from synthetic to real data.

Rather than classical fully connected layers before the Softmax as in Reference [30], we use a Global Average Pooling layer. According to Reference [41], there are two advantages to using GAP instead of fully connected layers. First, we have categories confidence maps at Conv 5 directly linked to each output class. Each map preserves the spatial information until the very last convolution layer. Secondly, the averaging operation before the Softmax will sum this spatial information across the map and produces an output that will be robust to small input variations, especially translations.

As a result, the last convolution layer, Conv 5, reduces the dimension of the output tensor of Conv 4 from 7×7×64 to 7×7×8. The output of Conv 5 is a tensor with eight feature maps, one per class. The Global Average Pooling operation then outputs an 8×1 vector from the output of Conv 5. This vector is the input of the Softmax layer to produce a probability distribution over each class. The final class prediction is given by the argument of the maximum of the Softmax output.

### 4.2. Training Consideration

We use the following experimental setting for training:The optimization algorithm used for training is Adam. We will use the sames configuration parameters as in Reference [45], i.e., $\alpha =0.001,\beta_{1}=0.9,\beta_{2}=0.999$ and $\epsilon =10^{-8}$.We use the Xavier initialization scheme [46] to prepare the network before training it during 150 epochs. To avoid over-fitting we use an early-stopping criterion to stop the learning process if the testing error starts to increase before reaching the last epoch.Use dropout [47] with a unit drop probability of 0.2, since we are only working with convolutions, as suggested in Srivatsava et al. [47].

This experimental setting results from cross-validation experiments.

## 5. Results

In this section, we report numerical experiments to evaluate the target recognition and identification performances for the proposed framework. We first report benchmarking experiments on transferring a model trained on synthetic data to real images with respect to other state-of-the-art schemes in Section 5.1. We then assess the robustness of target identification in Section 5.2, focusing on localization errors (Section 5.2.1) and scaling errors (Section 5.2.2).

### 5.1. CNN Performance on Real Data for Identification and Recognition

We first evaluate the recognition and identification performances for a perfect detection setting, i.e., when the patches provided as input of the target recognition and identification model relate to the true bounding box of the target. For benchmarking purposes, we compare the performance of our cfCNN with 130 thousand parameters to the following CNNs:the CNN from Rodgers et al. [30], with 310 thousand parameters.a Bilinear CNN [48], with both branches based on VGG16 [49] and 285 millions parameters.an Inception v3 [50], with 23 millions parameters.cfCNN(fc), a modified version of the cfCNN, where the convolution layers from Conv 1.1 to Conv 4 are kept. A fully connected layer with 256 units replaces Conv 5 and the GAP layer. Thus, this architecture has 540 thousand parameters.

For all experiments, we train a given architecture a minimum of 10 times and keep the best result on the considered validation dataset. In all cases, the training is performed on a synthetic dataset (TRS1, TRS2, or TRS3) and the performance is evaluated on real dataset DR.

CNNs training and testing were conducted on a workstation equipped with a Nivdia Tesla P4 GPU with 8 gigabytes of video memory. We report the following training time per batch:Rodgers et al. [30] with batch size 256: on average, 59 s per batch.BCNN [48] with batch size 96: on average, 251 s per batch.Inception v3 [50] with batch size 256: on average, 85 s per batch.cfCNN with batch size 256: on average, 55 s per batch.cfCNN(fc) with batch size 256: on average, 58 s per batch.

The SVM was trained on a 6 core Xeon E5-2680 v3. On the largest dataset (i.e., TRS3 with 18,944 images), the training took 4 h and 32 min.

We evaluate two different scores:Identification score: this score refers to the ability of the network to correctly predict the vehicle class, i.e., if its Top1- prediction is the correct label. It is the default global classification score.Recognition score: this score measures the ability of the network to correctly retrieve as the top-1 result a vehicle category similar to the one of the true label. For example if the result is c1 instead of being classified as c3, this contributes to the recognition score as it is from the same “category” as the expected result. This recognition score is more relevant than classically considered Top-3 or Top-5 scores for the targeted operational context.

#### 5.1.1. Performance Gains Using Improved Simulation

We first evaluate the impact of using more accurate synthetic datasets during training to improve identification and recognition results on real images for each architecture.

Table 2 shows the performance changes when training on TRS2 and TRS3 compared with the performance issued from a training on TRS1. The performance is assessed from the identification and recognition scores evaluated on dataset DR. For all networks and the SVM, we observe a better performance when improving the quality of the simulation data.

Overall, the largest gains for the recognition score are obtained by Inception v3 (+25.77%) and cfCNN (+17.98%). When considering the identification scores, both versions of the cfCNN provide the highest results. The high gains observed for the cfCNN(fc) can be explained by very low identification and recognition scores on the DR dataset after a training phase on TRS1. For the other architectures, we think that this does not reflect the performance of the network but the improvement brought by the increase in simulation quality.

#### 5.1.2. CNN Performance Comparison

We further compare the different architectures using Reference [30] as a baseline in Table 3. Generic computer vision architectures, namely BCNN and Inception V3, do not bring improvement compared with Reference [30], except in terms of the recognition score when training on dataset TRS3 (resp. +3.38% and +7.82%). Interestingly, the proposed cfCNN reaches the best performance in all training configurations, and the performance gain increases with the quality of the simulation data (e.g., from +11.25% in the identification score when training on dataset TRS1 to +19.54% when training on dataset TRS3). Improving the quality of the simulation has helped increase the recognition score of the SVM, up to 9% compared to Reference [30]. However, its performance on the identification task is below that of Reference [30]. The performance of the cfCNN(fc) compared to the cfCNN also highlights the benefits of GAP for transfer learning. Without GAP, the cfCNN(fc) managed to improve over Rodgers et al., the BCNN, and Inception v3 only on TS3 with an identification score of 8.10% and a recognition of 6.35%. Despite this improvement, its performance is still below the cfCNN with GAP. One can also note that the cfCNN gives the best results for identification purposes. These performance improvements also support the usefulness of realistic simulation datasets for learning-based strategies. These results further emphasize that the cfCNN takes the greatest benefit from improved simulation data.

#### 5.1.3. Gains Using Leaky ReLU as the Main Nonlinearity

As mentioned in Section 4.1, we used the leaky ReLU nonlinearity. To illustrate this effect, we consider two architectures: A cfCNN with the ReLU activation and a standard cfCNN with the leaky ReLU activation. Both networks were trained 10 times each on the TRS3 dataset and then tested on the DR dataset. When we compared the results form the best performing iteration of each network, we noted that the leaky-ReLU improved the performance of the cfCNN on the DR dataset. We reported a 4.75% increase in the identification score and a 2.87% in the recognition score. Additionally, the use of Leaky ReLu has allowed for a reduction of the standard deviation of identification and recognition results from each successive iteration from 8.24% and 5.23% respectively to 2.82% and 1.93% respectively.

#### 5.1.4. Dataset Size and Batch Size

The dataset and batch sizes have an impact on the classification performance. We experimented with smaller training datasets created from TRS3 and different batch sizes to see how it would affect identification and recognition results on the DR dataset.

Figure 7 illustrates the influence of the batch size used during training on the cfCNN performance on the DR dataset. Only the largest batch size values above 256 samples per batch have a strong negative impact on performance. A batch size of 256 exemplars provides a good balance between the identification and recognition performances.

Figure 8 illustrates the influence of the size of the training dataset on the cfCNN performance on the DR dataset. The horizontal axis values correspond to the fraction of the TRS3 dataset used during training. As expected, reducing the number of samples in the training set negatively impacts the performance on the real dataset.

### 5.2. Evaluation of the Robustness to Localization and Scaling Errors

In this section, we evaluate the robustness of target identification models w.r.t. the target localization and scaling errors. We compare the proposed cfCNN to the three CNN architectures previously used in Section 5.1. We also include in these experiments a SVM-based classifier, which is among the state-of-the-art shallow models for target classification in MIWR images.

In the following, the training step is based on dataset TRS3. For the evaluation of the robustness to viewpoint changes, we consider the two test datasets described in Section 3.3, namely
TTS to evaluate the robustness against localization errors. The results are presented in Section 5.2.1.TSS to evaluate the robustness against scaling errors. Test results is presented in Section 5.2.2.

We may stress that we do not apply any data augmentation during the training stage, such that trained models have never seen translated or scaled exemplars prior to the validation step. For each configuration, we retain the best model out of 10 training runs in terms of global identification performance (i.e., the identification score) for centered samples.

#### 5.2.1. Impact of Localization Errors on Identification Performance

We report in Figure 9 the comparison of the identification performance under localization errors for the benchmarked models for vertical and horizontal localization errors. All models reach very similar performances for ideal cases (with no localization error as shown in Figure 9). Standard CNN models, namely BCNN, Inception v3, and Rodgers et al., do not significantly improve over the SVM model. When using either the BCNN or Inception v3, an identification performance is only 20% better than the SVM at most. For a translation factor above 0.2, the identification performance is still below 70% in all directions for both networks.

By contrast, the proposed cfCNN clearly outperforms all other models, including the SVM. It is the only model which keeps an identification score above 80% for a localization error up to 30% of the bounding box. Other models only maintain such a performance for a localization error up to 10% of the bounding box. The benefits of the GAP layer is also shown when comparing the score of the cfCNN against the cfCNN(fc), of which the performance is similar to the other CNNs.

Figure 10 shows the identification score obtained with cfCNN for each target. It can be noted that the behavior of the results is not always symmetric. This can be easily explained by the infrared signature of the targets which is not necessarily symmetric. For example for a battle tank, as depicted in Figure 11, cells such as the exhaust plume or the commander hatch clearly induce asymmetry. At some points, the translations can also lead to occlusions of some parts of the targets. This masking effect is typically stronger for a translation along the horizontal axis.

#### 5.2.2. Robustness against Scale Modification

We report below the robustness performance regarding scaling effects. Figure 12 provides the results obtained on dataset TSS. The proposed cfCNN leads to better results when the bounding box scale reduces compared to its original size. However, SVM has a slight advantage over the cfCNN when the bounding box scale increases. It appears that the identification score is always higher for the cfCNN compared to the other architectures. Again, the cfCNN(fc) identification scores are below the cfCNN with GAP.

One can also note that when the bounding box size is reduced (i.e., with a scale factor below 1), some parts of the targets may be masked. Thus, these results may also be interpreted in terms of robustness to masking effects.

## 6. Conclusions

In this paper, we have introduced a fully convolutional CNN architecture with global average pooling for target recognition and identification in infrared images. We highlighted the need of realistic synthetic images during the training step in order to reach a good recognition performance for real scenes.

Our experiments clearly demonstrate that the cfCNN outperforms state-of-the-art architectures. Especially, the cfCNN keeps identification scores above 70% for localization and scale errors in all directions. Experimentally, the use of a Global Average Pooling layer was shown to be a key feature of our architecture to reach this performance.

Future work will further investigate recognition tasks in the presence of partial occlusions (e.g., the presence of buildings, trees, etc.) as well as novelty detection issues.

## Figures and Tables

**Figure 1 sensors-19-02040-f001:**
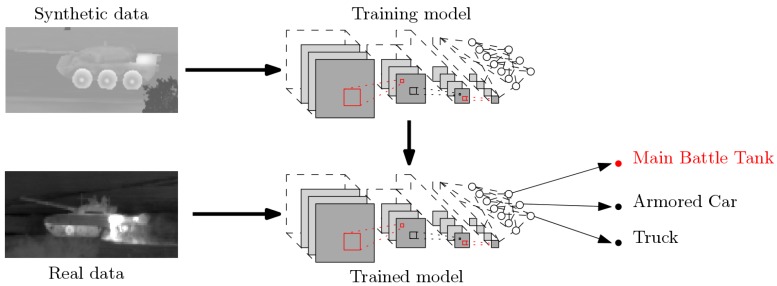
A schematic depicting the proposed approach for target identification and recognition: a convolutional neural network (CNN) model trained using simulated data (image generated with OKTAL-SE) and tested on real data (image extracted from SENSIAC database [8]).

**Figure 2 sensors-19-02040-f002:**
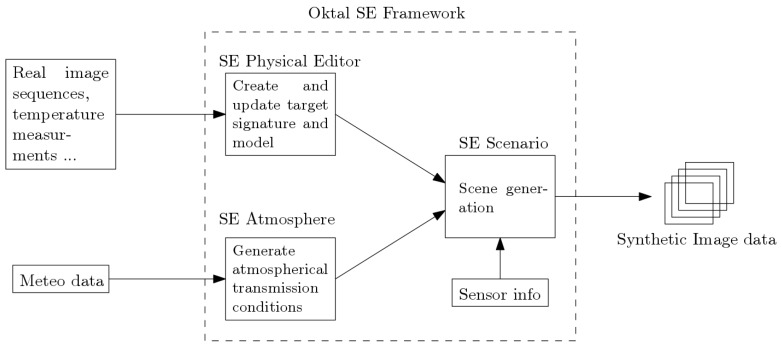
A simplified workflow description of the creation of a synthetic IR scene using OKTAL-SE).

**Figure 3 sensors-19-02040-f003:**
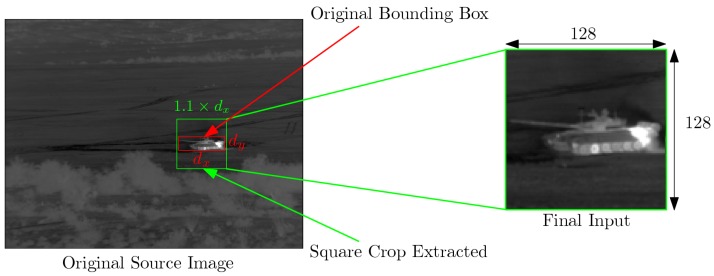
An extracted patch image (in green) from the original infrared image using the original bounding box information (in red).

**Figure 4 sensors-19-02040-f004:**
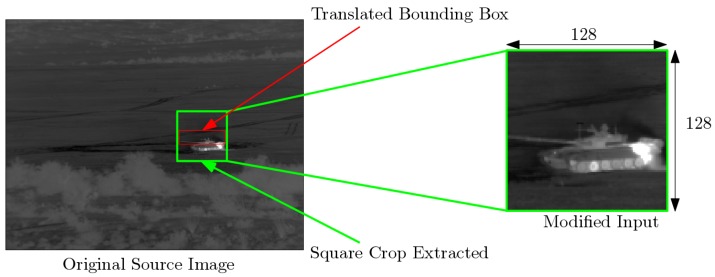
An extraction of target exemplars accounting for localization errors: We consider here an upward translation by a factor of 0.3 of the height of the bounding box; we depict both the original bounding box (red) and the translated bounding box (green).

**Figure 5 sensors-19-02040-f005:**
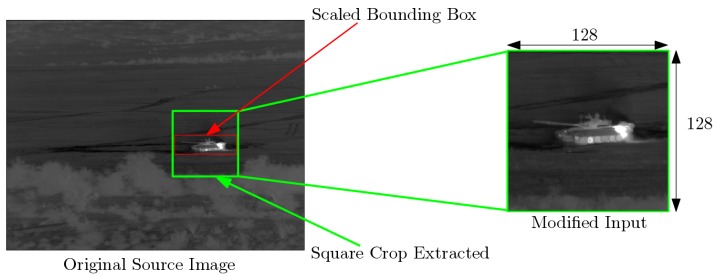
An extraction of target exemplars accounting for scaling errors: The original bounding box is shown in red, and the scaled one is in green.

**Figure 6 sensors-19-02040-f006:**
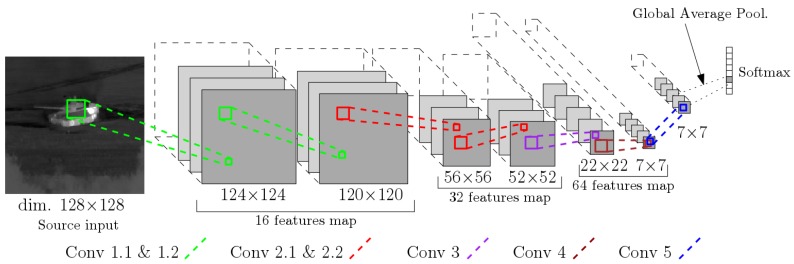
A compact and fully convolutional neural network used for infrared target recognition and identification.

**Figure 7 sensors-19-02040-f007:**
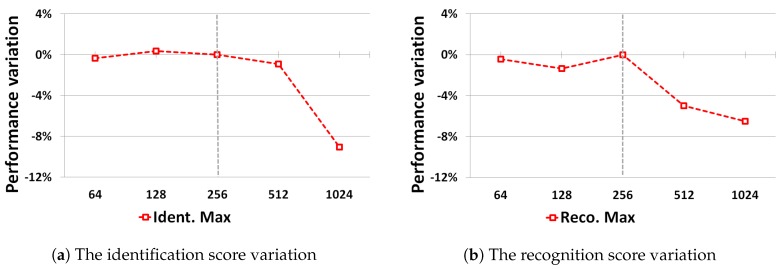
The influence of batch size on the best identification and recognition results on the DR dataset.

**Figure 8 sensors-19-02040-f008:**
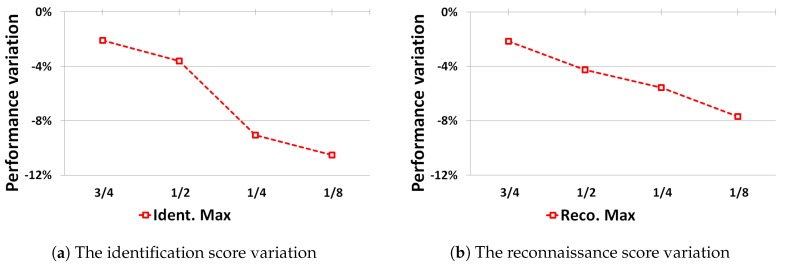
The influence of the dataset TRS3 size used during training on the best identification and reconnaissance results on the DR.

**Figure 9 sensors-19-02040-f009:**
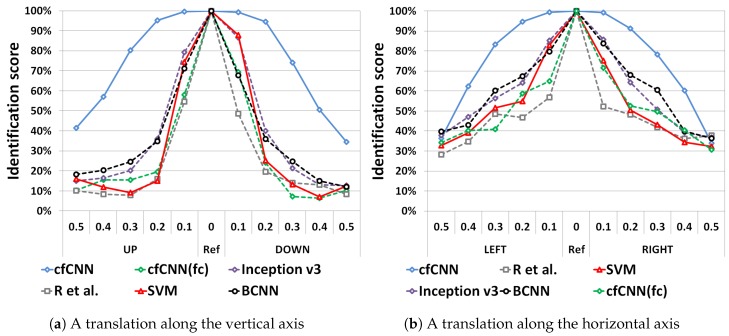
The global identification results for translated examples on the TTS dataset.

**Figure 10 sensors-19-02040-f010:**
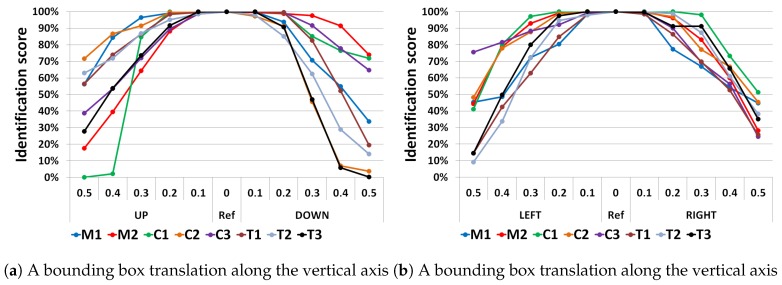
The identification scores obtained using the cfCNN architecture when translating the bounding box. M(i), C(i), and T(i) refer to each target used in the dataset.

**Figure 11 sensors-19-02040-f011:**
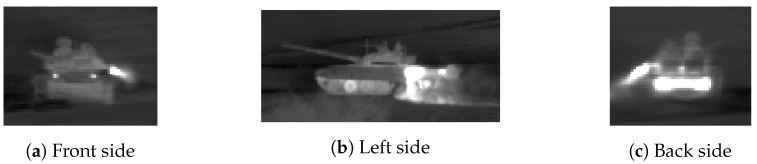
Examples of infrared signatures of a T72 main battle tank with regard to its orientation.

**Figure 12 sensors-19-02040-f012:**
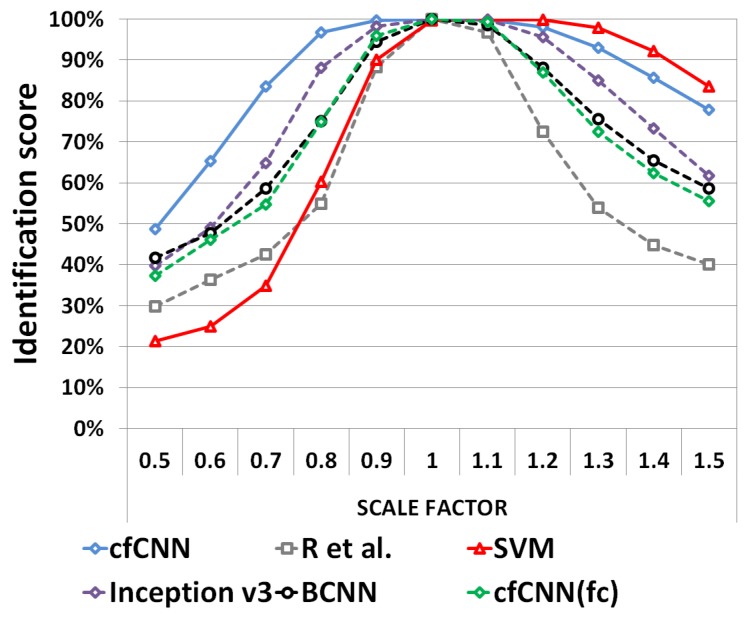
The identification scores obtained on the TSS dataset.

**Table 1 sensors-19-02040-t001:** Details of the proposed architecture.

Layer Name	Output Features Map	Kernel Size	Output Size
Conv 1.1	16	5 × 5	124 × 124
Conv 1.2	16	5 × 5	120 × 120
Max-pool 2 × 2			60 × 60
Conv 2.1	32	5 × 5	56 × 56
Conv 2.2	32	5 × 5	52 × 52
Max-pool 2 × 2			26 × 26
Conv 3	64	5 × 5	22 × 22
Max-pool 2 × 2			11 × 11
Conv 4	64	5 × 5	7 × 7
Conv 5	8	1 × 1	7 × 7
Global average pooling			8 × 1
Softmax			8 × 1

**Table 2 sensors-19-02040-t002:** The performance gains of different architectures on the real dataset (DR) relative to the same architecture trained on the TRS1 dataset. Each column indicates on which dataset the network was trained: TRS2 or TRS3. This include the results for the recognition and identification tasks.

Training Set	TRS2		TRS3	
Architecture	Ident.	Recog.	Ident.	Recog.
SVM	3.2%	10.20%	8.8%	16.11%
Rodgers et al.	3.34%	8.36%	8.43%	9.27%
BCNN	3.38%	14%	4.48%	12.64%
Inception v3	0.79%	10.68%	9%	25.77%
cfCNN(fc)	14.17%	12.71%	23.61%	17.71%
cfCNN	6.20%	10.62%	14.67%	17.98%

**Table 3 sensors-19-02040-t003:** A performance evolution of different network architectures on the DR dataset relative to the architecture of Rodgers et al. Includes results for the recognition and identification tasks.

Training Set	TRS1		TRS2		TRS3	
Architecture	Ident.	Recog.	Ident.	Recog.	Ident.	Recog.
SVM	−1.44%	−1.78%	−2.96%	2.47%	−1.92%	9%
BCNN	−0.13%	−8.91%	−0.05%	−0.40%	−6.29%	3.38%
Inception v3	−2.47%	−15.83%	−9.35%	−10.98%	−1.25%	7.82%
cfCNN(fc)	−18.02%	−5.43%	1.42%	0.98%	8.10%	6.35%
cfCNN	11.25%	3.31%	15.48%	5.96%	19.54%	14.38%

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
