# Peer review of "CNN-Based Target Recognition and Identification for Infrared Imaging in Defense Systems"

_sensors, 2019, doi:10.3390/s19092040_

Reviewer 1 Report

This paper proposes a CNN structures for detected objects in defence systems. The paper uses known techniques for object detection while the challenges that a defence dataset may cause is not clearly described. Some major questions are:

1) In the introduction, I would like to see the differences and the challenges that the new content may cause. For instance, there are papers for background subtraction in thermal imaging Why don't you use them instead of this approach?

Makantasis, Konstantinos, et al. "Data-driven background subtraction algorithm for in-camera acceleration in thermal imagery." IEEE Transactions on Circuits and Systems for Video Technology 28.9 (2018): 2090-2104.

2) Section 2 should be extended to include references on deep learning and computer vision such as 

Voulodimos, Athanasios, et al. "Deep learning for computer vision: a brief review." Computational intelligence and neuroscience 2018 (2018).

Längkvist, M., Karlsson, L., & Loutfi, A. (2014). A review of unsupervised feature learning and deep learning for time-series modeling. Pattern Recognition Letters42, 11-24.

Guo, Yanming, Yu Liu, Ard Oerlemans, Songyang Lao, Song Wu, and Michael S. Lew. "Deep learning for visual understanding: A review." Neurocomputing 187 (2016): 27-48

3) Section 24 should go to Section 1 and should be extended a lot.

4) the computation cost for testing and training should be revealed

5) How do the parameters of the network affect the whole process

6) Please compare the performance with other shallow learning techniques.

7) Please compare the performance with other non-machine learning object detection techniques in thermal imaging. 

8) How does the number of samples affect the performance? Please give curves.

Author Response

This paper proposes a CNN structures for detected objects in defence systems. The paper uses known techniques for object detection while the challenges that a defence dataset may cause is not clearly described.

We would like to thank the reviewer for the feedback on our manuscript. We have revised section 1 and 2 to better point out the challenges that come from applying our techniques to IR imaging for defence-oriented applications. All the modifications in the paper are highlighted in red.

1. In the introduction, I would like to see the differences and the challenges that the new content may cause. For instance, there are papers for background subtraction in thermal imaging Why don't you use them instead of this approach?

Makantasis, Konstantinos, et al. "Data-driven background subtraction algorithm for in-camera acceleration in thermal imagery." IEEE Transactions on Circuits and Systems for Video Technology 28.9 (2018): 2090-2104.

The robustness to novelty and background changes is of interest in the field. However, we believe that using background suppression techniques might lead to masking effects for some target as we may encounter cold and stationary targets that can blend with the background. This point might affect the results we show in Section 5.2. We discuss this issue in Section 2.1 in the revised manuscript.

Especially, the above reference exploits video sequences with successive frames to improve background suppression. In the considered case-study, we only consider still images for an extensive range of environments and vehicle state. We believe the analysis of IR video streams to be clearly of interest for future work, especially for the improvement of the detection stage, we assume to be part of a pre-processing step in the current manuscript.

2. Section 2 should be extended to include references on deep learning and computer vision such as:

Voulodimos, Athanasios, et al. "Deep learning for computer vision: a brief review." Computational intelligence and neuroscience 2018 (2018).

Längkvist, M., Karlsson, L., & Loutfi, A. (2014). A review of unsupervised feature learning and deep learning for time-series modeling. Pattern Recognition Letters, 42, 11-24.

Guo, Yanming, Yu Liu, Ard Oerlemans, Songyang Lao, Song Wu, and Michael S. Lew. "Deep learning for visual understanding: A review." Neurocomputing 187 (2016): 27-48

Thank you for your suggestions. As we do not address video analysis, we have included the first and third references in the revised manuscript in Section 2. We also expanded Section 2.1 with a paragraph to link our work and the use of CNNs in automatic target recognition with computer vision and deep learning literature.

3. Section 24 should go to Section 1 and should be extended a lot.

Following this suggestion, we have revised the last paragraph of Section 1 to better state the specific positioning of our paper in the introduction Section.

4.the computation cost for testing and training should be revealed

The revised manuscript clarifies the relative computational complexity of the benchmarked models in terms of number of parameters and of the processing time using our implementation using Tensorflow 1.5.0 and a workstation with a Tesla P4 GPU in section 5.1.

5. How do the parameters of the network affect the whole process

Training parameters were chosen mostly to improve training time without affecting classification performance of the cfCNN on the real images. Small variations in Batch size and learning rate around the values we selected had a limited impact on performance. More details on batch size influence have been added to section 5.1.4.

We noted that the use of leaky ReLU instead of ReLU as our main non-linearity improved the performance on the real dataset. We modified our manuscript and added Section 5.1.3 to present the results.

6. Please compare the performance with other shallow learning techniques.

Following this suggestion and the one from Reviewer 2, while not lowering the readability of the reported numerical experiments, we have complemented results in Section 5.1.1 and 5.1.2 using the SVM based approach used in section 5.2 and we added results using our cfCNN but without GAP to illustrate its effect. We might also point out that we regard the experiments reported with  the SVM (using Histogram of Oriented Gradients as feature extractor) as an upper-bound of the performance of a classic shallow approach. We hope these new results will be satisfactory.     

7. Please compare the performance with other non-machine learning object detection techniques in thermal imaging.

In this paper, we do not deal with the detection part. We assume that a given detector provides images, each one containing a unique target. In Section 5., we demonstrate the robustness of the proposed framework to target localization errors, which is regarded as a key contribution.

8. How does the number of samples affect the performance? Please give curves.

 We have expanded section 5.1 and added sub-section 5.1.4 with curves in Fig. 8 and 9. showing the impact of dataset size and batch size on cfCNN performance.

Reviewer 2 Report

comments:

1. This paper applied fully connected CNN in object identification and recognition in the wild for infrared imaging in defense applications,which is very prospective in this filed.

2. The authors basically explained their method clearly. The paper is fluent in language and clear in expression.

3. Authors introduced how to prepare training and testing data in section 3. Both of them are cropped from the original images. So for the original images, how could you find out the location of

the patch?

4. It's better to give more details about how to get simulation data.

5. In the conclusion, authors concluded the use of a Global Average Pooling layer was shown to be a key feature of the proposed architecture.It's better to give more detailed information and analysis about

this.Such as replacing the Global Average Pooling layer with fully connected layer and then comparing their results. 

Author Response

1&2. This paper applied fully connected CNN in object identification and recognition in the wild for infrared imaging in defense applications, which is very prospective in this field.

The authors basically explained their method clearly. The paper is fluent in language and clear in expression.

We would like to thank the reviewer for the positive feedback on the manuscript and its contents. Responses to the comments and corrections are detailed below. They are highlighted in red in the updated manuscript.

3. Authors introduced how to   prepare training and testing data in section 3. Both of them are cropped from   the original images. So for the original images, how could you find out the   location of the patch?

For simulated datasets i.e. DS1 DS2 and DS3, the target position and bounding box is added as metadata for each image during its creation. This aspect is briefly mentioned at the beginning of section 3.2. Regarding real images (i.e. DR), the bounding boxes were manually annotated for each image. We have modified the manuscript accordingly to clarify these points.

4. It's better to give more details about how to get simulation data.

Unfortunately, the considered simulated data are not open-source data. We further detail the Section 3.1 and 3.3 in the revised manuscript the simulation process and the differences between the three simulation datasets according to the confidential constraints.

The two new paragraphs are accompanied by a figure (fig. 3 in the revised manuscript) briefly outlining the database creation process.

5. In the conclusion, authors concluded the use of a Global Average Pooling layer was shown to be a key feature of the proposed architecture. It’s better to give more detailed information and analysis about this. Such as replacing the Global Average Pooling layer with fully connected layer and then comparing their results.

We report additional experiments using a GAP or fully-connected layer as final layer of the proposed architecture. These experiments show that, for the cfCNN, the addition of a GAP layer significantly improves the performance on the real dataset and on the translated or scaled inputs.

We have revised our manuscript accordingly. It now includes these complementary experiments (Tab. 2 and Tab. 3, p. 9), which are presented in Section 5.1, and 5.2. The new tables now include results from a cfCNN where the GAP layer has been replaced by a fully connected layer.

Reviewer 3 Report

This paper has introduced a fully convolutional CNN architecture with global average pooling for target recognition and identification in infrared images, and highlighted the need of realistic synthetic images during the training step in order to reach good recognition performance for real scenes. The effectiveness of the proposed method has been amply confirmed by the experiments. So, I think the paper can been published with some minor revisions:

1. Much more details about the simulated and real infrared data should been given, just like reference 30. For readers, they may interested very much in the influence of the accuracy of simulation on the recognition and identification performance, maybe you can think about making your experiment data sets public to all.

2. Additional references about CNN-based SAR image target identification and recognition should been added, such as:

[1] S. Chen, H. Wang, F. Xu, and Y. Q. Jin, “Target classification using the deep convolutional networks for SAR images,” IEEE Trans. Geosci. Remote Sens., vol. 54, no. 8, pp. 4806–4817, Aug. 2016.

[2] Z. Lin, K. F. Ji, M. Kang, X. G. Leng and H. X. Zou, “Deep convolutional highway unit network for SAR target classification with limited labeled training data,” IEEE Trans. Geosci. Remote. Sens. Lett., vol. 14, no. 7, pp. 1091-1095, Jul. 2017.

[3] Z. Huang, Z. Pan, B. Lei, “Transfer learning with deep convolutional  neural network for SAR target classification with limited labeled data,” Remote Sensing, vol. 9, no. 9, pp. 907, 2017.

[4] Kang Miao, Ji Kefeng, Leng Xiangguang etc., “Synthetic Aperture Radar Target Recognition with Feature Fusion Based on a Stacked Autoencoder," Sensors, 17(1), 2017.01.

Author Response

This paper   has introduced a fully convolutional CNN architecture with global average   pooling for target recognition and identification in infrared images, and   highlighted the need of realistic synthetic images during the training step   in order to reach good recognition performance for real scenes. The   effectiveness of the proposed method has been amply confirmed by the   experiments. So, I think the paper can been published with some minor   revisions

The authors would like to thank the reviewer for the positive feedback on the article. The responses to each comment is detailed below along with the proposed modifications to the manuscript (highlighted in red) if applicable.

1. Much more details about the simulated and real infrared data should been given, just like reference 30. For readers, they may interested very much in the influence of the accuracy of simulation on the recognition and identification performance, maybe you can think about making your experiment data sets public to all.

Unfortunately, the considered dataset cannot be made open-source. We have revised Section 2.2 and 3.3 to improve the description of simulated datasets (according to the confidential constraints), especially their differences in terms of simulation hypotheses.

The two new paragraphs are accompanied by a figure (fig. 3 in the revised manuscript) briefly outlining the database creation process.

2. Additional   references about CNN-based SAR image target identification and recognition   should been added, such as:

[1] S. Chen,   H. Wang, F. Xu, and Y. Q. Jin, “Target classification using the deep convolutional networks for SAR   images,” IEEE Trans. Geosci. Remote Sens., vol. 54, no. 8, pp. 4806–4817,   Aug. 2016.

[2] Z. Lin,   K. F. Ji, M. Kang, X. G. Leng and H. X. Zou, “Deep convolutional highway unit   network for SAR target classification with limited labeled training data,”   IEEE Trans. Geosci. Remote. Sens. Lett., vol. 14, no. 7, pp. 1091-1095, Jul.   2017.

[3] Z. Huang,   Z. Pan, B. Lei, “Transfer learning with deep convolutional  neural   network for SAR target classification with limited labeled data,” Remote   Sensing, vol. 9, no. 9, pp. 907, 2017.

[4] Kang   Miao, Ji Kefeng, Leng Xiangguang etc., “Synthetic Aperture Radar Target   Recognition with Feature Fusion Based on a Stacked Autoencoder,"   Sensors, 17(1), 2017.01.

Thank you for the proposed additional references. As these references address SAR imaging, we have included the most relevant ones, in terms of methodological positioning for our work addressing MWIR imaging, in section 2.1 and 2.2 of the revised manuscript.

Round  2

Reviewer 1 Report

Some of my comments should be addressed again.

1)

The robustness to novelty and background changes is of interest in the field. However, we believe that using background suppression techniques might lead to masking effects for some target as we may encounter cold and stationary targets that can blend with the background. This point might affect the results we show in Section 5.2. We discuss this issue in Section 2.1 in the revised manuscript. 

Especially, the above reference exploits video sequences with successive frames to improve background suppression. In the considered case-study, we only consider still images for an extensive range of environments and vehicle state. We believe the analysis of IR video streams to be clearly of interest for future work, especially for the improvement of the detection stage, we assume to be part of a pre-processing step in the current manuscript.

This paragraph and the respective citation should be somehow added to the document.

2) I know that this paper does not deal with image analysis. But a well explained and concrete discussion for SURE increases the readability of the paper. 

3)

5. How do the parameters of the network affect the whole process

Training parameters were chosen mostly to improve training time without affecting classification performance of the cfCNN on the real images. Small variations in Batch size and learning rate around the values we selected had a limited impact on performance. More details on batch size influence have been added to section 5.1.4.

We noted that the use of leaky ReLU instead of ReLU as our main non-linearity improved the performance on the real dataset. We modified our manuscript and added Section 5.1.3 to present the results.

4) Please compare the performance with other non-machine learning object detection techniques in thermal imaging.

In this paper, we do not deal with the detection part. We assume that a given detector provides images, each one containing a unique target. In Section 5., we demonstrate the robustness of the proposed framework to target localization errors, which is regarded as a key contribution.

The comment here was to compare the method with other data analysis and statistical processing techniques, and not techniques for detection. Please revise. 

Please put some figures to show this effect.

4)

Author Response

We would like to thank the reviewer for the new feedback on our manuscript. Unfortunately, the format of the new comments we received (reproduced in what follows) make sometime unclear some of your pending. We are really sorry about that. The previous responses are in red, the comments in black and our responses to the new comments are in blue, as well as in the paper.

Some of my comments should be addressed again.

1)

The robustness to novelty and background changes is of interest in the field. However, we believe that using background suppression techniques might lead to masking effects for some target as we may encounter cold and stationary targets that can blend with the background. This point might affect the results we show in Section 5.2. We discuss this issue in Section 2.1 in the revised manuscript.

Especially, the above reference exploits video sequences with successive frames to improve background suppression. In the considered case-study, we only consider still images for an extensive range of environments and vehicle state. We believe the analysis of IR video streams to be clearly of interest for future work, especially for the improvement of the detection stage, we assume to be part of a pre-processing step in the current manuscript.

This paragraph and the respective citation should be somehow added to the document.

We added a new section section 2.1 in the manuscript based on our previous answer. We also added references to target detection and recognition techniques in IR that do not rely on deep learning to better contextualise our article.

2) I know that this paper does not deal with image analysis. But a well explained and concrete discussion for SURE increases the readability of the paper.

3)

Is there any question about this point?

5. How do the parameters of the network affect the whole process

Training parameters were chosen mostly to improve training time without affecting classification performance of the cfCNN on the real images. Small variations in Batch size and learning rate around the values we selected had a limited impact on performance. More details on batch size influence have been added to section 5.1.4.

We noted that the use of leaky ReLU instead of ReLU as our main non-linearity improved the performance on the real dataset. We modified our manuscript and added Section 5.1.3 to present the results.

Once again, we are sorry, but is our previous response about this point adequate for you?

4) Please compare the performance with other non-machine learning object detection

techniques in thermal imaging.

In this paper, we do not deal with the detection part. We assume that a given detector provides images, each one containing a unique target. In Section 5., we demonstrate the robustness of the proposed framework to target localization errors, which is regarded as a key contribution.

The comment here was to compare the method with other data analysis and statistical processing techniques, and not techniques for detection. Please revise.

In our previous answer to comment n°6 we mention that we consider the SVM as an upper bound in term of classification performance for shallow techniques. Using a combination of HOG, PCA and three different classifiers we have obtained the following results:

Training set

TRS2

TRS3

Ident

Reco

Ident

Reco

Classification Tree

-5.21%

-5.62%

-7.31%

-3.08%

Naïve Bayes

7.61%

6.42%

9.36%

12.29%

KNN

1.74%

6.49%

3.46%

1.96%

Table 1: Performance gains of different architectures on the DR dataset relative to the same architecture trained on the TRS1 dataset. Each column indicate on which dataset the network was trained: TRS2 or TRS3. Include results for the recognition and identification tasks.

As for the results presented in section 5.1 the table above shows that improving the quality of the simulation improve classification performance except for our classification tree.

TRS1

TRS2

TRS3

Algorithm

Ident.

Reco.

Ident.

Reco.

Ident.

Reco.

Class. Tree

-20.72%

4.51%

-34.08%

-14.62%

-42.61%

-19.69%

Naïve Bayes

-44.14%

-12.43%

-34.38%

-13.63%

-36.87%

-6.94%

KNN

-16.09%

2.25%

-17.90%

-0.49%

-19.02%

-0.90%

SVM

-1.44%

-1.78%

-2.96%

2.47%

-1.92%

9%

Table 2: Performance evolution of different networks architectures on the DR dataset relative to the architecture of Rodgers et al. Include results for the recognition and identification tasks.

The table 2 above shows that indeed we can consider the SVM as an upper bound for our use-case when compared to other machine learning approaches. We believe that adding the results to table 2 and 3 may affect the clarity of section 5. Thus, we choose not to include them in the manuscript.

As mentioned before, we modified section 2 to include references to non-deep learning techniques for ATR for the reader.

Please put some figures to show this effect.

Once again, we are really sorry, but at this point, we do not understand what effect you would like us to illustrate with figures?

4)

Is there any question about this point?

Reviewer 2 Report

The author made a revision to the paper based on my opinion and answered all my questions.

But I still have some comments and hope the author can further explore in this topic. The author demonstrated the GAP (global average pooling) has greater advantages than fully connected layer in the problem issued in their work . In addition to numerical results, maybe the author can also analyze it from other aspects, such as the features within the CNN networks, to give readers more insightful information.

Author Response

The author made a revision to the paper based on my opinion and answered all my questions.

But I still have some comments and hope the author can further explore in this topic. The author demonstrated the GAP (global average pooling) has greater advantages than fully connected layer in the problem issued in their work. In addition to numerical results, maybe the author can also analyze it from other aspects, such as the features within the CNN networks, to give readers more insightful information.

We would like to thank the reviewer for the new feedback on our manuscript. Unfortunately, we cannot provide a sample of the feature maps with our test dataset for confidentiality reasons.

For the networks with and without GAP, the feature maps up to Conv 4 will be very similar. However, with GAP we use 2D feature maps (Conv 5) instead of  a fully connected layer. We do not have developed techniques to compare the 2D feature maps to their equivalent in the fully connected layer.

Some intuitive insight on the potential advantages of GAP is available in the original article from Lin et al. We see two benefits to the use of GAP in our network. The first is the direct correspondence between a feature map and an output class directly from the start of the training. They will ultimately represent “categories confidence maps” and preserve the spatial information up until the very last layer of the network. The second benefit is the averaging operation, which sums this spatial information and produce an output that will be robust to small translations of the input.

When using fully connected layers we lose both aspects. Direct correspondence between units from the last layer is not present and by flattening the feature maps from conv 4 we also lose the spatial information.

We added a paragraph in section 4.1 to explain our choice of GAP and introduce the elements explained above in the manuscript.

Round  3

Reviewer 1 Report

All my comments have been addressed.